# Antibodies to full-length and the DBL5 domain of VAR2CSA in pregnant women after long-term implementation of intermittent preventive treatment in Etoudi, Cameroon

Jean Claude Djontu[1,2]*, Yukie Michelle Lloyd[3], Rosette Megnekou[1,2], Reine Medouen Ndeumou Seumko'o[1,2], Ali Salanti[4,5], Diane Wallace Taylor[3], Rose Gana Fomban Leke[2]

1 Department of Animal Biology and Physiology, Faculty of Science, University of Yaoundé I, Yaoundé, Cameroon, 2 The Biotechnology Center, University of Yaoundé I, Yaoundé, Cameroon, 3 John A. Burns School of Medicine, University of Hawaii at Manoa, Honolulu, Hawaii, United States of America, 4 Department of Immunology and Microbiology, Center for Medical Parasitology, University of Copenhagen, Copenhagen, Denmark, 5 Department of Infectious Diseases, Copenhagen University Hospital, Copenhagen, Denmark

* cdjontu@yahoo.fr

## Abstract

In high malaria transmission settings, the use of sulfadoxine-pyrimethamine-based intermittent preventive treatment during pregnancy (IPTp-SP) has resulted in decreased antibody (Ab) levels to VAR2CSA. However, information of Ab levels in areas of low or intermediate malaria transmission after long-term implementation of IPTp-SP is still lacking. The present study sought to evaluate antibody prevalence and levels in women at delivery in Etoudi, a peri-urban area in the capital of Yaoundé, Cameroon, that is a relatively low-malaria transmission area. Peripheral plasma samples from 130 pregnant women were collected at delivery and tested for IgG to the full-length recombinant VAR2CSA (FV2) and its most immunogenic subdomain, DBL5. The study was conducted between 2013 and 2015, approximately ten years after implementation of IPTp-SP in Cameroon. About 8.6% of the women attending the clinic had placental malaria (PM). One, two or 3 doses of SP did not impact significantly on either the percentage of women with Ab to FV2 and DBL5 or Ab levels in Ab-positive women compared to women not taking SP. The prevalence of Ab to FV2 and DBL5 was only 36.9% and 36.1%, respectively. Surprisingly, among women who had PM at delivery, only 61.5% and 57.7% had Ab to FV2 and DBL5, respectively, with only 52.9% and 47.1% in PM-positive paucigravidae and 77.7% of multigravidae having Ab to both antigens. These results suggest that long-term implementation of IPTp-SP in a low-malaria transmission area results in few women having Ab to VAR2CSA.

## Introduction

In pregnant women, *Plasmodium falciparum*-infected erythrocytes (IE) express an antigen, VAR2CSA, that participates in the binding of IE to chondroitin sulfate A (CSA) on the syncytiotrophoblast lining the intervillous space of the placenta [1, 2]. The sequestration of IE in

**Data Availability Statement:** All relevant data are within the manuscript and its supporting information files.

**Funding:** The World Academy of Sciences (TWAS) received by RM, supported this work; research Grant No: 12-081 RG/BIO/AF: AC_I—UNESCO FR: 3240271366, Faculty of Science, University of Yaoundé I. The Luminex MAGpix was provided by grant P30GM11473, Centers of Biomedical Research Excellence, National Institute of General Medical Sciences, NIH; received by DWT research team. The funders had no role in study design, data collection and analysis, decision to publish, or preparation of the manuscript.

**Competing interests:** The authors have declared that no competing interest exist.

placental tissue results in the pathogenesis of placental malaria (PM). PM is an important risk factor for maternal anemia and delivery of low birth weight babies [3, 4], which remains one of the leading causes of mortality and morbidity in neonates and infants in sub-Saharan Africa. Thus, preventing malaria in pregnant women is necessary not only to reduce maternal morbidity but also to minimize the consequence of PM on the health of their offspring. Upscaling of malaria control strategies, particularly effective case management with the use of rapid diagnostic tests, use of intermittent preventive treatment with sulfadoxine-pyrimethamine during pregnancy (IPTp-SP), and insecticide-treated bed nets (ITN), have led to a worldwide reduction of malaria-related morbidity and mortality [5]. However, compliance with intervention strategies together with the expanding resistance to anti-malarial drugs and insecticides pose a great challenge. In addition, SP cannot be administered in women during the first trimester of the pregnancy, although *P. falciparum* infection is also frequent and harmful during this period [6]. The development of an effective vaccine against PM may offer a sustainable solution to protect mothers and their babies from malaria-related morbidity and mortality in endemic areas.

As the major *P. falciparum* surface protein that mediates IE accumulation in the placenta [7, 8], leading to inflammation, the VAR2CSA antigen is the main target of protective immunity to PM. In many malaria endemic areas, exposure to malaria is perennial and pregnant women acquire antibodies (Ab) to VAR2CSA over successive pregnancies that improve pregnancy outcomes [9, 10]. Previous studies reported that parasites with a VAR2CSA knock-out gene irreversibly lose the ability to adhere to CSA [11], and that Ab to VAR2CSA inhibit IE binding to CSA *in vitro* [12, 13]. This evidence strongly supports VAR2CSA as the leading candidate for a PM vaccine. The rationale for a PM vaccine is to induce immunological memory to the IE with the VAR2CSA phenotype, and elicit an accelerated response upon subsequent *P. falciparum* infections. Prior studies reported that high malaria transmission is required for natural acquisition of long-lasting Ab response to PM [14]. Thus, any variable that affects the risk for *P. falciparum* exposure could influence the magnitude and the quality of Ab response to the potential VAR2CSA-based vaccine. Although SP inhibits folic acid synthesis in malaria parasites, which is required for parasite replication, decreasing thereby the exposure of pregnant women to *P. falciparum* antigens, conflicting data exist about the impact of IPT on Ab response to VAR2CSA. In fact, the use of IPTp-SP has resulted in decreased Ab levels to VAR2CSA in some high transmission settings [15, 16], while no impact of the treatment on the Ab response was found in other lower malaria endemic areas [17,18]. This suggest that the impact of IPT on Ab response to VAR2CSA changes across different geographic conditions.

The eco-epidemiological profile of malaria transmission in Cameroon is very heterogenic and is made of three patterns (Sahelian, soudanian and equatorial), different by their malaria transmission period and entomological indices [19]. Within a given pattern, some variation is also observed; for example, in the equatorial pattern where this study was carried out, the transmission increases from the center urban to rural areas. Available data on Ab response to VAR2CSA in Cameroonian women exist only for central urban (low malaria transmission) [20, 21] or rural area (high malaria transmission) [14, 15, 22, 23], although these areas are usually separated by a peri-urban or sub-urban area where malaria transmission is intermediate or relatively low. In addition, most of previous studies on Ab response to VAR2CSA in Cameroon were carried out before the IPTp-SP implementation. The present study sought to evaluate the prevalence of anti-VAR2CSA (FV2 and DBL5) Ab and levels of Ab in women at delivery in Etoudi, a peri-urban area in the capital of Yaoundé, Cameroon, that is a relatively low-malaria transmission area, after long-term implementation of IPTp-SP. The results of the study will help to improve the design of future clinical trials in malaria endemic areas, on the efficacy of the potential VAR2CSA-based vaccine against PM.

## Materials and methods

### Ethical considerations

The study protocol was reviewed and approved by the National Ethics Committee of Cameroon (Ethical Clearance 2013/02/ N˚ 029/L/CNERSH/SP). Administrative Authorizations were obtained from the Ministry of the Public Health of Cameroon (No D30-392 AAR/MIN-SANTE/SG/DROS/ CRC/ CEA1) and from Health Centre. Participation in the study was voluntary with written informed consent from each woman. Upon collection, all samples and clinical information about the mother and baby were de-identified. The study was performed following the guidelines and regulations of human clinical research as recommended by the Ministry of the Public Health of Cameroon. Malaria rapid diagnostic test (RDT) was performed for each woman at the time of enrollment and positive results were reported to the physician for prescription of treatment according to the national policy.

### Study population and sample collection

This cross-sectional study was carried out between 2013 and 2015 at the Marie Reine Health Center in Etoudi, situated in a peri-urban area of Yaoundé, Cameroon. Malaria transmission in this area is relatively low and perennial, peaking in May (during the long-wet season from March to June) and October (during short wet season from September to November). A total of 130 HIV-negative women aged 16 to 39 years were recruited. Information on the mother's health, estimated length of pregnancy, parity, age, use of anti-malarial drugs, IPTp-SP usage, HIV status, and baby birth weight were recorded. Peripheral blood samples were collected in EDTA tubes from women immediately following delivery. A portion of the blood was used to prepare thick and thin smears for microscopy and to measure hemoglobin levels. The remainder was centrifuged and plasma was collected and stored at -80º C for antibody studies. Placental tissues were also collected and a section excised to prepare impression smears and for histology.

### Diagnosis of placental malaria and determination of hemoglobin levels

Thick and thin blood smears, were prepared using Giemsa-Wright stain and read by two skilled microscopists to determine the presence of malaria parasites. In addition, thin film of peripheral blood was used to determine parasite species. Results from blood smears were compared with RDT results obtained at enrollment using the Carestart[TM]HRP2 (Pf) (Access Bio Inc. NJ, USA). Placental sections were fixed in buffered formalin, embedded, stained with hematoxylin-eosin, and examined for IE and malaria pigments. Women were considered PM + if IE were detected in impression smears of villous tissue and/or in histological sections. Hemoglobin levels in maternal blood were determined using a Coulter Counter (URIT-3300, Europe).

### Measurement of antibodies

Plasma IgG Ab levels to VAR2CSA recombinant proteins (FV2 and DBL5) of the FCR3 strain, expressed in Baculovirus-transfected insect cells, were measured using a multi-analyte platform (MAP). The coupling of the recombinant proteins to MagPix microspheres and basic protocol have been described previously [21, 24, 25]. Briefly, 50 μl of antigen-coupled microspheres (2000 microspheres/antigen) were incubated in a well of a microtiter plate (U-bottom microplate) with 50 μl of plasma diluted to 1:100 in phosphate buffered saline containing 1% bovine serum albumin (PBS 1% BSA) for 1 h at room temperature on a rotating shaker at 500 rpm (Microplate Shaker, Lab-line, Melrose Park, IL, USA). After washing twice with PBS-

0.05% Tween 20 and once with PBS-1% BSA, 100 μl of secondary Ab (R-phycoerythrin-conjugated, Affini Pure F(ab′)2 fragment, Goat anti-human IgG Fc fragment specific, Jackson Immunoresearch, West Grove, PA, USA) diluted to 2 μg/ml in PBS-1% BSA was added to each well and incubated on a shaker in the dark for 1 h. Wells were washed and microspheres re-suspended in 100 μl PBS-1% BSA. Plates were analysed immediately using MAGpix *Analyser (MAGpix Technology, USA)*. The reader was programmed to read a minimum of 100 beads per spectral address, DD Gate 7500–15,000 and 35s timeout. The results were expressed as median fluorescence intensity (MFI). Positive and negative controls were included on each plate to control for plate-to-plate variation. Positive control was a pool of eight Cameroonian multigravidae with high Ab levels to VAR2CSA. Negative controls were plasma from 12 Cameroonian men, age 23 to 30 years of age, living in the study area. The cut-off value for seropositivity was determined based on the mean MFI + 2SD of the male samples.

## Statistical analysis

Graph Pad Prism 6.0.1 was used for the statistical analyses. Continuous variables are reported as means +/- standard deviations (SD) or medians with interquartile range (IQR). Differences between groups were compared using unpaired t-test for normally distributed continuous data or Mann-Whitney Rank Sum test for non-normal distributed continuous data, while categorical variables were reported as percentages and were compared using Fisher's exact test. P values <0.05 were considered statistically significant.

# Results

## Study population

The characteristics of women in this study are summarized in Table 1. Initially,105 women were enrolled between June 2013 and February 2014, among whom 9 women had PM, providing an estimated prevalence of PM of 8.6% (9/105). Then, between May and June 2014, 25 women were enrolled to collect additional samples from PM+ women. Among the 25

**Table 1. Characteristics of the study groups.**

| | PM+ Women (n = 26) | PM- Women (n = 104) | P values* |
|---|---|---|---|
| Age in years (mean ±SD) | 24.8 ± 5.2 | 26.5 ± 5.2 | 0.1311 |
| Gravidity (mean ±SD) | 2.6 ± 1.7 | 2.9 ± 1.7 | 0.3622 |
| Parity (mean ±SD) | 2.1 ± 1.1 | 2.5 ± 1.3 | 0.1770 |
| Maternal hemoglobin in (g/dL) (mean ±SD) | 10.6 ± 1.5 | 12.8 ± 1.18 | <0.0001 |
| Hematocrit (mean % PCV ±SD) | 31.8 ± 4.9 | 38.7 ± 3.6 | <0.0001 |
| Percentage (%) of women with anemia | 57.7 | 3.9 | <0.0001 |
| Peripheral parasitemia: IE/μl [median and 25%-75% IQR] | 2889 [177, 66365] | 0 | |
| Placental parasitemia (%) [median and 25%-75%IQR] | 2.6 [0.1, 7.6] | 0 | |
| Length of gestation (mean weeks ± SD) | 39.2 ± 2.7 | 39.9 ± 2.36 | 0.3769 |
| Baby birth weight (mean g ± SD) | 3060 ± 471 | 3387 ± 526 | 0.0046 |
| Percentage (%) of LBW babies | 11.5 | 3.9 | 0.1425 |
| Number of Doses of SP (mean ± SD) | 1.6 ± 1.2 | 2.2 ± 1.1 | 0.0101 |
| Percentage (%) women using ITNs** | 57.7 | 77.7 | 0.0482 |

*Analysis by PM status included: Student t test for normally distributed means; Mann Whitney for non-parametric medians (PCV); and Fisher's Exact test for comparisons of proportions.

** Percentage of women who reported using insecticide treated bednets (ITNs). LBW: low birth weight; SP: sulfadoxine-pyrimethamine.

additional women, 17 were confirmed to have PM+, providing a total of 26 PM+ and 104 PM-negative women in the study (total 130 pregnant women) (Table 1). Among the 26 PM+ women, only 80% were peripheral blood-smear positive for malaria. The PM+ women tended to be younger and pauciparous compared to PM- women, but their mean age, gravidity and parity were not significantly different (age [24.8 vs 26.5 years, p = 0.13]; gravidity [2.6 vs 2.9, p = 0.36] and parity [2.1 v 2.5, p = 0.18]) (Table 1). However, maternal hemoglobin levels and hematocrits were significantly lower in PM+ compared to PM- women (hemoglobin 10.6 vs 12.8 g/dl and hematocrit 31.8 vs 38.7%, with both p values <0.0001) and the percentage of women with anemia was higher in the PM+ group (57.7% vs 3.9%, p<0.0001). Among the 26 women with PM, a wide range in placental parasitemia was found (median parasitemia: 2.5% with interquartile range of 0.1% to 7.6%). The mean baby birth weight in PM+ women was significantly lower than that of PM- women (3,060 vs 3,387 g, p = 0.0046); although the percentage of low birth weight babies did not differ significantly between the groups (11.5 vs 3.9%, p = 0.14). The mean number of IPTp-SP doses taken by PM+ women was lower than PM- women (1.6 vs 2.2 doses, p = 0.01) and the proportion of women using bed nets was lower in the PM+ group compared to PM- group (57.7 vs. 77.7%, p = 0.048). A comparison between women who do not use IPT during pregnancy (n = 22) with women who took 3 doses (n = 70) showed that women receiving 3 doses of SP were older (23.5 vs 26.5 years, p = 0.014), had higher hemoglobin levels (11.8 g/dL vs. 12.8, p = 0.0048) and PCV (35.5% vs. 38.6%, p = 0.005), and had fewer cases of placental malaria (31.8% vs. 5.8%, p = 0.0426).

## Levels and prevalence of IgG Ab to VAR2CSA in PM+ and PM- women

Ab levels and the percentage of PM+ and PM- women who had Ab toVAR2CSA antigens were investigated (Fig 1). For the 130 women, Ab levels to FV2 and DBL5 were higher in PM+ women than PM- women (p = 0.0059 and 0.002, respectively) (Fig 1A and 1B). Furthermore, when Ab levels were evaluated for only those women who were Ab-positive for FV2 and DBL5, Ab levels remained significantly higher in the PM+ than PM- group (p = 0.0019 and 0.0079, respectively) (Fig 1C). At term, 61.5% and 57.7% of PM+ women had Ab to FV2 and DBL5, respectively; whereas, only about half as many (31.7% and 30.8%) of PM- women had Ab to these antigens (Table 2). Thus, high Ab levels were associated with infection and many women lacked Ab to the VAR2CSA antigens.

## Levels and prevalence of IgG Ab to VAR2CSA in women by gravidity status

Among PM+ women, the levels of Ab to FV2 and DBL5 did not differ significantly between paucigravid and multigravid women (median: FV2: 10,459 vs 21,078 MFI p = 0.19; DBL5: 3,030 vs 24,450 MFI p = 0.10 (Fig 2). In addition, there was no difference in the prevalence of Ab to FV2 (52.9% vs 77.8%, p = 0.40) or DBL5 (47.1% vs 77.8% p = 0.22). However, there was a significant difference among PM- women, with multigravidae (MG) having significant higher median Ab levels to FV2 and DBL5 (3,240 and 2,333 MFI) than paucigravidae (1,330 and 1,261 MFI respectively) (both p values ≤0.0001). Similarly, the percentage of PM- MG women having Ab to FV2 and DBL5 (48.2% and 44.6%) was significantly higher than that of paucigaviadae (12.5% and 14.6%, respectively) (p = 0.0001 and 0.0003, respectively) (Table 2).

## Influence of use of SP on Ab levels and prevalence

The impact of IPTp-SP on Ab levels and percentage of women who were sero-positive at delivery for FV2 and DBL5 was assessed (Fig 3, Table 3). The use of two or three doses of SP did not significantly impact either the percentage of women with Ab to FV2 and DBL5 or Ab levels in Ab-positive women compared to women not taking or taking only one dose of SP.

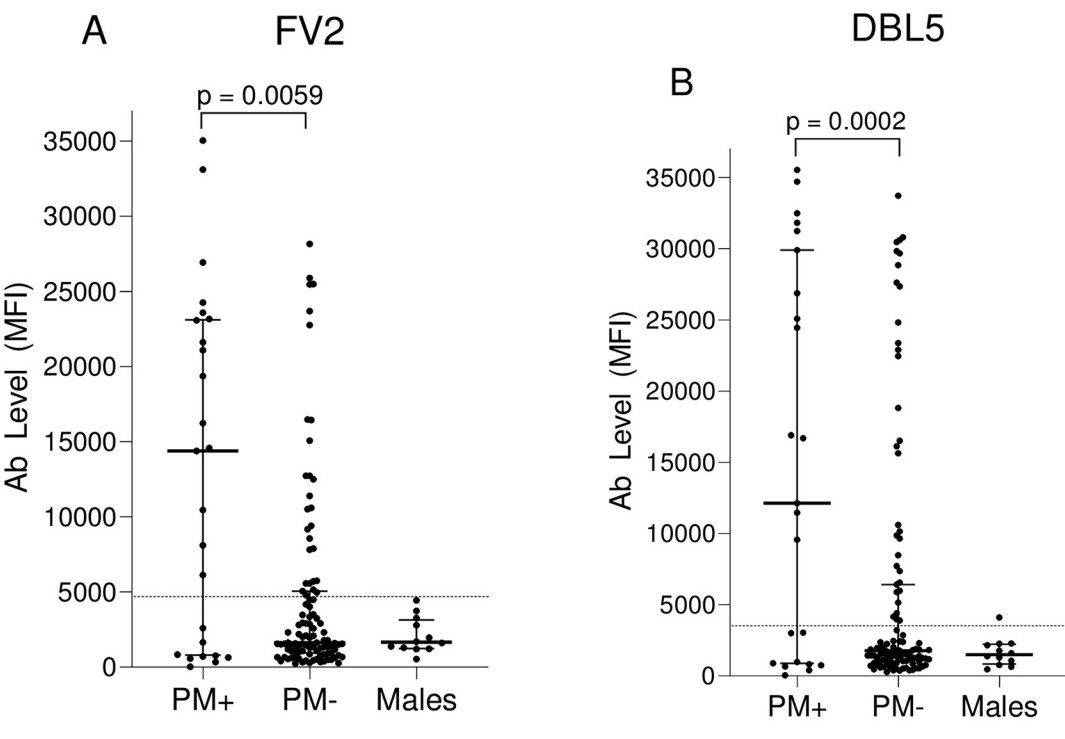

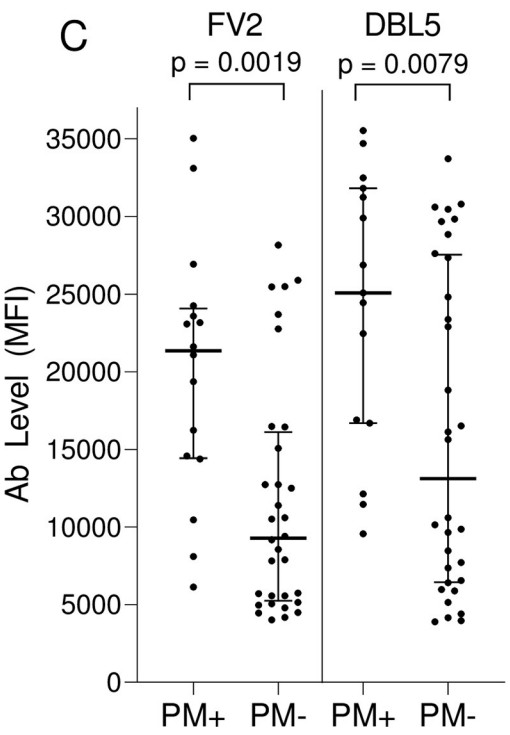

**Fig 1. Antibody levels to VAR2CSA (FV2 and DBL5) in PM+ and PM+ women.** Plasma Ab levels to VAR2CSA were compared in women by PM status using Mann-Whitney Rank Sum test. A and B: IgG levels for all women (n = 130); C: IgG levels for women who were Ab-positive (i.e., had levels higher than the cut-off (n = 49 women for FV2, n = 47 for DBL5); MFI: median fluorescence intensity; IQR: interquartile range; Ab: antibodies; PM: placental malaria. The horizonal dotted lines show the cut-off for Ab-positive, which was 4,483 MFI for FV2 and 3,232 MFI for DBL-5. Horizontal bars represent the group median +/- 25% IQR.

**Table 2. Percentage of women with antibodies to VAR2CSA (FV2 and DBL5) at delivery by PM and gravidity status.**

|  | PM+ | | | PM- | | P value* |
|---|---|---|---|---|---|---|
| FV2 | 16/26 (61.5%) | | | 33/104 (31.7%) | | 0.0067 |
| DBL5 | 15/26 (57.7%) | | | 32/104 (30.8%) | | 0.0131 |
|  | Pauci | MG | P value* | Pauci | MG | P value* |
| FV2 | 9/17 (52.9%) | 7/9 (77.8%) | 0.3989 | 6/48 (12.5%) | 27/56 (48.2%) | 0.0001 |
| DBL5 | 8/17 (47.1%) | 7/9 (77.8%) | 0.2167 | 7/48 (14.6%) | 25/56 (44.6%) | 0.0003 |

*Fisher exact test was used for comparison between paucigravidae and multigravidae. The results are presented as percentage. The cut-off for Ab-positive was 4,483 MFI for FV2 and 3,232 MFI for DBL5. Pauci: paucigravidae; MG: multigravidae

## Discussion

The present study sought to evaluate Ab prevalence and levels to VAR2CSA (FV2 and DBL5) in women at delivery in Etoudi, a peri-urban area in Yaoundé, Cameroon, that is a relatively low-malaria transmission area, after long-term implementation of IPTp-SP. The transmission of malaria is decreasing in many parts of the world and Ab levels to *P. falciparum* antigens may decline in the absence of boosting. Studies have reported that Ab levels to VAR2CSA decreased in high transmission areas [15, 16] or remained unchanged in lower transmission areas in women receiving IPT [17, 18]. Thus, investigations in areas with different malaria transmission intensities are required, because the impact of IPT on the Ab responses changes under different geographic conditions.

IPTp-SP has been implemented in Cameroon since 2004 and the coverage rate of IPTp-SP (at least one dose of SP during pregnancy) is currently estimated at 30% in the rural areas and 70% in the urban areas [25]. In the current study, 83% of women took at least one dose of IPTp-SP and 74% reported sleeping under a bed net during pregnancy, suggesting high compliance to these preventive measures. Although it is recommended that all Cameroonian women take at least three doses of SP during pregnancy (starting from the second trimester) [26], SP is not free in some private health centers and is not prescribed for women taking some drugs, such as cotrimoxazole. These reasons help explain why about 17% of women in this study did not take SP. Compared to women who do not use IPTp-SP during pregnancy, women taking 3 doses were older, had higher hemoglobin levels and PCV, and had fewer cases of PM. Thus, IPT-SP is having a positive impact in Etoudi. This finding is in line with that of a previous study that have reported an association between IPTp-SP usage and decrease of cases of PM in Cameroonian women [15]. However, a small proportion (8/70) of women who took the complete three-dose regimen of IPTp-SP still had *P. falciparum* infection. This could be due to the presence of resistance to the drug as it has been reported in Cameroon [27], differences in the duration of SP half-life in the individual, or timing of treatment. Encouraging pregnant women and clinicians to adhere to the new WHO policy that recommends the use of SP at each prenatal visit after the first trimester [28] could further reduce the prevalence of placental malaria.

As expected, few women had Ab to FV2 and DBL5 in this low transmission area, where IPTp-SP intervention has been used for >10 years. In fact, among the 104 PM- women at delivery only 31.7% and 30.8% had Ab to FV2 and DBL5 respectively, and only 12.5% of paucigravidae and 48% of MG had Ab to FV2 (Table 2). Although pre-IPT Ab prevalence in Etoudi is unknown, Ab prevalence to FV2 in PM- women living in Yaoundé prior to implementation was 24.3% in pauci- and 40.1% in MG [20]. Thus, with decreased transmission and extended use of IPT, today fewer women have Ab to FV2 during their first two pregnancies, but Ab prevalence is similar in MG. Since the Ab response was higher in PM+ than PM- women (Fig

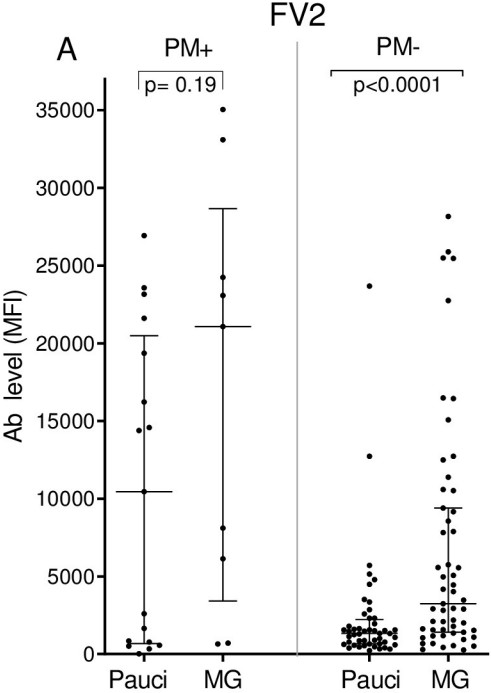

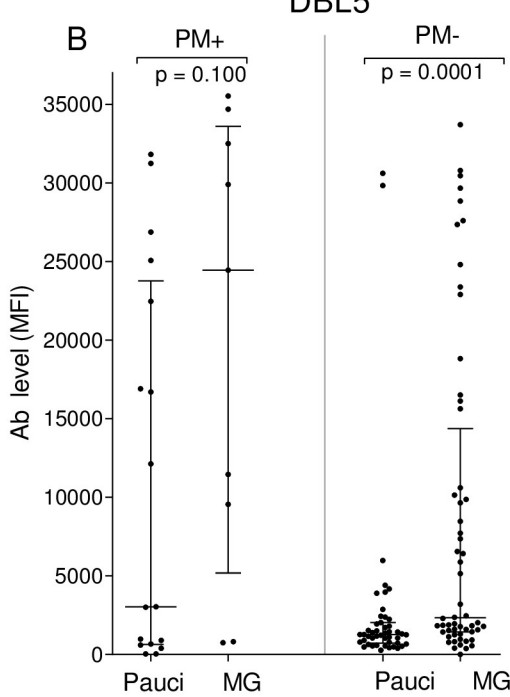

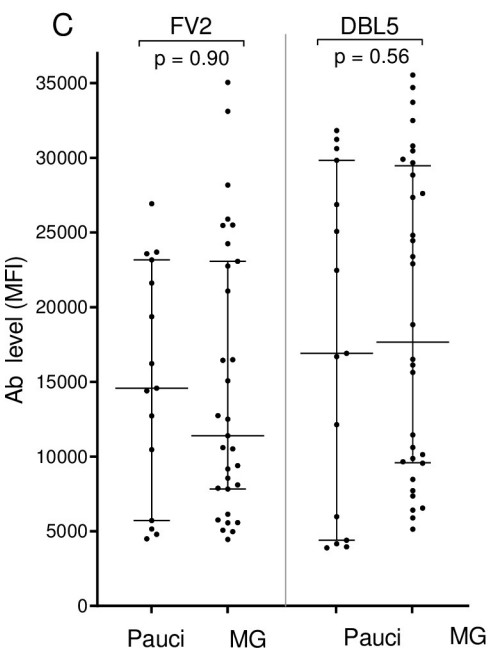

**Fig 2. Antibody levels to VAR2CSA (FV2 and DBL5) in paucigravid and multigravid women.** Plasma levels of Ab to VAR2CSA were compared between paucigravidae and multigravidae using the Mann-Whitney Rank Sum test. A and B: IgG Ab levels for all women (n = 130); C: IgG levels for women with Ab levels higher than the cut-off (i.e., results for only Ab+ women) (n = 49 women for FV2, n = 47 women for DBL5); MFI: median fluorescent intensity; IQR: interquartile range; Pauci: Paucigravid women; MG: Multigravid women; Ab: antibody; Cut-off for Ab-positive was 4,483 MFI for FV2 and 3,232 MFI for DBL5. Horizontal bars represent the group median +/- 25% IQR.

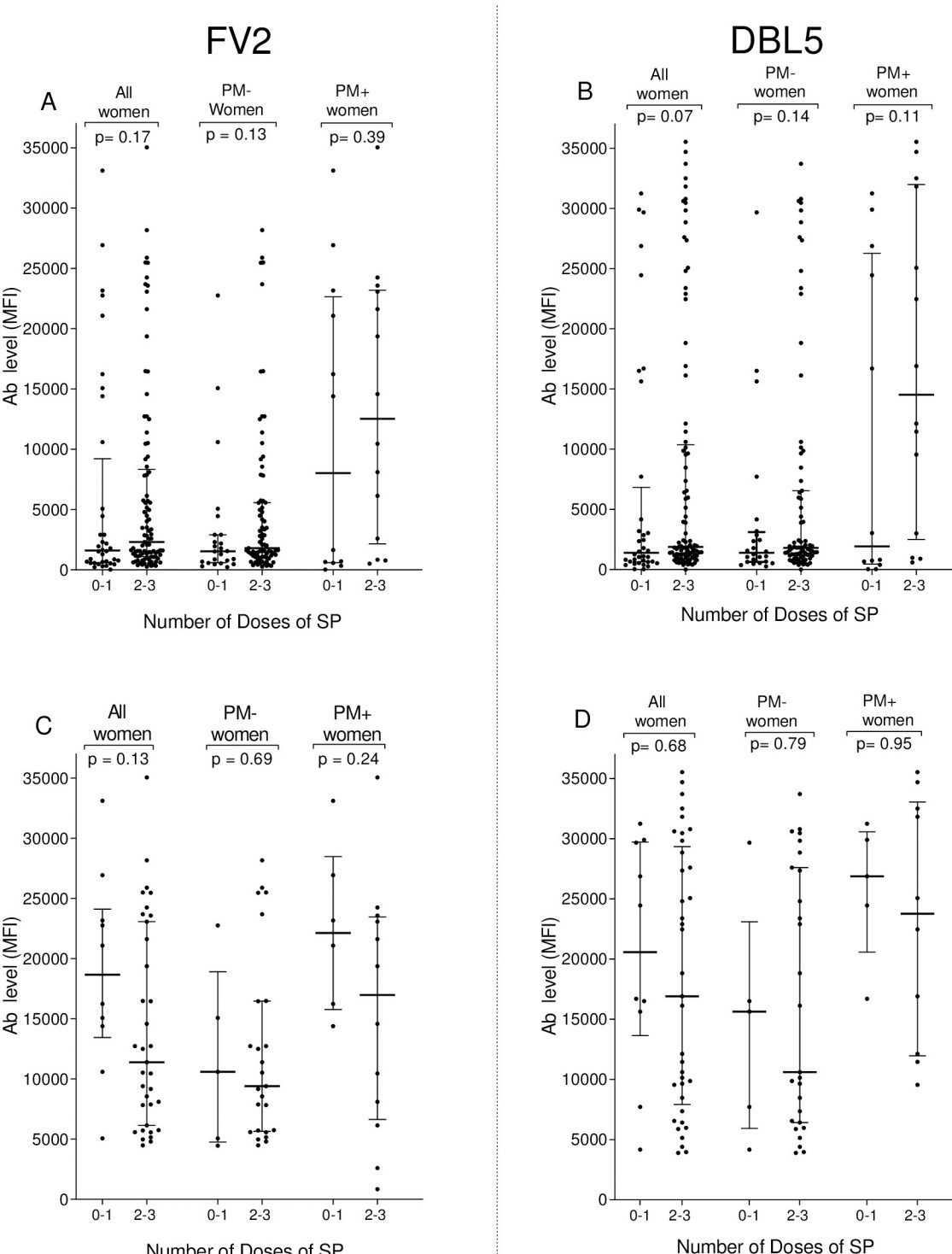

**Fig 3. Influence of number of doses of SP on Ab levels to VAR2CSA (FV2 and DBL5) at delivery.** Mann-Whitney Rank Sum test was used to compare median plasma Ab levels to VAR2CSA between groups. A and B: IgG levels for all women (n = 130). C and D: IgG levels for Ab-positive women (n = 45 women for FV2, n = 47 women for DBL5). MFI: median fluorescence intensity; IQR: interquartile range; Ab: antibody; The cut-off for Ab-positivity was 4,483 MFI for FV2 and 3,232 MFI for DBL5. Horizontal bars represent the group median +/- 25% IQR.

**Table 3. Percentage of women with antibodies to VAR2CSA (FV2 and DBL5 FCR3) at delivery by use of IPTp-SP.**

| | 0–1 Dose | 2–3 Doses | p value* |
|---|---|---|---|
| | n = 36 | n = 94 | |
| FV2 | 10/36 (27.8%) | 35/94 (37.2%) | 0.4104 |
| DBL-5 | 10/36 (27.8%) | 37/94 (39.4%) | 0.3377 |

*Comparison between women who received 0–1 dose and 2–3 doses of SP. The results are presented as percentage. Fisher exact test was used. The cut-off for Ab-positivity was 4,483 MFI for FV2 and 3,232 MFI for DBL5.

1), Ab to these pregnancy-associated antigens are a marker of infection and not protection. One of the more relevant findings of this study is that Ab prevalence was similar in paucigravidae (1 to 2 pregnancies) and MG (3 or more pregnancies) who have PM to FV2 (52.9% vs 77.8%, p = 0.40) or DBL5 (47.1% vs 77.8% p = 0.22). Thus, even when parasites were sequestered in the placenta, many women lacked Ab to VAR2CSA at delivery. In addition, the difference in Ab levels between pauci- and multi-gravidae was not significantly different (median: FV2: 10,459 vs 21,078 MFI p = 0.19; DBL5: 3,030 vs 24,450 MFI p = 0.10, respectively (Fig 2). The combined data suggest that some MG women had not become infected during previous pregnancies due to low malaria transmission and the use of IPT-SP and produced either a primary or weak secondary Ab response during the current pregnancy; whereas, other MG produced higher levels of Ab (Fig 2), although the difference was neither statistically significant nor robust enough to eliminate IE sequestered from the placenta. The absence of Ab in low transmission areas suggests that VAR2CSA is not highly immunogenic and that i) repeated exposure is needed to induce a strong Ab response to VAR2CSA and/or ii) Ab to other malarial antigens help control parasitemia [20]. Since few women in Etoudi have high Ab levels to VAR2CSA, a VAR2CSA vaccine would be beneficial not only to primigravidae, but to multigravidae as well.

In this study, over 38% of women who had PM failed to produce Ab to FV2 and DBL5 (Table 2). Since IE were sequestered in the placenta, the question becomes, why didn't they produce detectable Ab to VAR2CSA? Several explanations seem feasible. First, some of the women may have become infected too close to delivery to have produced Ab. However, since malaria transmission was perennial and over one-third of PM+ women lacked Ab, one would expect most women to be infected prior to the last 2–3 weeks of pregnancy. Second, the SP may have reduced or eliminated malarial parasites to levels below that required to induce an immune response. However, since Ab levels in women receiving 0–1 doses versus 2–3 doses had similar Ab levels (Fig 3) and prevalence (Table 3), this explanation does not entirely answer the question. Thirdly, IPTp-SP could result in very low, persistent levels of VAR2CSA that induce T-reg responses. Fourthly, in this study, only the FcR3 allelic form of VAR2CSA was used. Since VAR2CSA is a polymporphic protein [29], it is possible that women have Ab to variants of FV2 or DBL5 that were not detected. Finally, multiple infections might be required to induce a sustainable immune response to VAR2CSA. In support of this possibility are data from a study conducted in Yaoundé (low transmission) and Ngali II (high transmission) [22]. In a cohort of women who are either slide- or PCR-positive for *P. falciparum* before 6 months of pregnancy, at delivery only ~32% primi- and 40% multi-gravidae in Yaoundé compared to 80% of primi- and >90% of multi-gravidae in Ngali II had Ab to FV2 at delivery [14]. Thus, multiple infections during pregnancy may be needed to boost the response, and the level of exposure is not sufficient in lower transmission areas or when IPTp-SP is used. No matter what the explanation, not all women who have PM make Ab to VAR2CSA. Determining why they don't, could be of value for designing a vaccine for preventing PM.

Many questions remain about the impact of IPT-SP on acquisition of immunity to PM. In the current study, PM was diagnosed by detecting infected erythrocytes in the intervillous space of placenta. Thus, some women in the PM- group could have had submicroscopic infections that might have boosted their Ab response [30]. Although Ab detected in this study recognized FV2 and DBL5, it is unclear if they are able to block the binding of IE to CSA. VAR2CSA is a polymorphic protein and the FcR3 variant used in this study may not be the predominant strain current circulating in Etoudi. Since functional Ab to placental parasites are variant dependent [31], a number of molecular, immunological and epidemiological studies are needed before a complete understanding of the role of VAR2CSA is determined and how IPT-SP alters the natural response to this important antigen.

## Conclusion

The results of this study suggest that long-term implementation of IPTp-SP in a relatively low-malaria transmission area results in few women having Ab to VAR2CSA and Ab levels remain low in those with and without PM.

## Supporting information

**S1 File.**
(XLSX)

## Acknowledgments

We thank all the members and staff of the Biotechnology Center Nkolbisson, University of Yaoundé I, Cameroon, for fieldwork and laboratory tests. Special thanks to Anna Babakhanyan for her contribution of reagents and mentorship. We thank the Marie Reine Health Centre in Etoudi, Cameroon, for allowing sample collection. We send our sincere gratitude to the women and men enrolled in the study for making this study possible.

## Author Contributions

**Conceptualization:** Jean Claude Djontu, Rosette Megnekou, Rose Gana Fomban Leke.

**Data curation:** Jean Claude Djontu, Yukie Michelle Lloyd, Rosette Megnekou, Diane Wallace Taylor.

**Formal analysis:** Jean Claude Djontu, Yukie Michelle Lloyd, Rosette Megnekou, Diane Wallace Taylor.

**Funding acquisition:** Rosette Megnekou, Diane Wallace Taylor.

**Investigation:** Jean Claude Djontu.

**Methodology:** Jean Claude Djontu, Reine Medouen Ndeumou Seumko'o.

**Project administration:** Jean Claude Djontu.

**Resources:** Rosette Megnekou, Ali Salanti, Rose Gana Fomban Leke.

**Software:** Jean Claude Djontu.

**Supervision:** Rosette Megnekou, Rose Gana Fomban Leke.

**Validation:** Diane Wallace Taylor, Rose Gana Fomban Leke.

**Visualization:** Jean Claude Djontu, Rose Gana Fomban Leke.

**Writing – original draft:** Jean Claude Djontu, Yukie Michelle Lloyd, Diane Wallace Taylor.

**Writing – review & editing:** Jean Claude Djontu, Yukie Michelle Lloyd, Diane Wallace Taylor.

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
