## [Decision Letter · Decision Letter 0]

9 Jun 2020

PONE-D-20-13928

Antibodies to full-length and the DBL5 domain of VAR2CSA in pregnant women after long-term implementation of intermittent preventive treatment in Etoudi, Cameroon

PLOS ONE

Dear Dr. DJONTU,

Thank you for submitting your manuscript to PLoS ONE. After careful consideration, we felt that your manuscript requires substantial revision, following which it can possibly be reconsidered, thus governing the decision of a “major revision”. As requested by the reviewers, the authors need to address several concerns, particularly related to the study design, methods and results. For example, the study would benefit from an additional interpretation of the data. Study limitation should be included (how about submicroscopic infections). We therefore invite you to submit a revised version of the manuscript paying close attention to the specific points raised by both reviewers.  For your guidance, a copy of the reviewers' comments was included below

We look forward to receiving your revised manuscript.

Kind regards,

Luzia Helena Carvalho, Ph.D.

Academic Editor

PLOS ONE

Journal Requirements:

Reviewers' comments:

Reviewer's Responses to Questions

**Comments to the Author**

1. Is the manuscript technically sound, and do the data support the conclusions?

Reviewer #1: Yes

Reviewer #2: Yes

2. Has the statistical analysis been performed appropriately and rigorously? 

Reviewer #1: Yes

Reviewer #2: Yes

3. Have the authors made all data underlying the findings in their manuscript fully available?

Reviewer #1: Yes

Reviewer #2: Yes

4. Is the manuscript presented in an intelligible fashion and written in standard English?

Reviewer #1: Yes

Reviewer #2: Yes

5. Review Comments to the Author

Reviewer #1: I enjoyed reading this well-written manuscript examining antibody responses to the pregnancy-specific malaria protein VAR2CSA in women in Cameroon. Understanding more about the impact of intermittent preventive treatment in pregnancy on population levels of antimalarial immunity is important for designing appropriate malaria interventions to achieve malaria elimination.

I have no major concerns with the scientific approach or conclusions drawn from the findings. I would recommend the following minor changes:

Line 60, page 3: Consider changing “efficient” vaccine to “effective” vaccine, but then you use effective twice, so perhaps change to the following, or similar:

“The development of an effective vaccine against PM may offer a sustainable solution to protect mothers and their babies from malaria-related morbidity and mortality in endemic areas”.

Line 138-139, page 7: The reference provided (Ref 24; Babakhanyan et al 2016) in turn refers to earlier papers for details on coupling of recombinant proteins to MagPix microspheres, so I would suggest including the reference to these earlier papers.

Line 162, page 8: Change “unpair t-test” to “unpaired t-test”. Also provide a general statement, similar to what is included in footnote to table 1, on when t-tests were used and when Mann-Whitney Rank Sum were used (e.g. normal versus nonnormal continuous data)

Line 168-170, page 8: Upon initial reading it was not clear what was meant by “among whom 105 were randomly enrolled between June 2013 and February 2014”. Does this mean that a subset of 105 were randomly selected from the total of 130, or that 105 was the number of women who were enrolled between June 2013 and February 2014 (with the remaining 25 enrolled outside this period)?. I continued to read, and this was clarified to some extent at line 171, but the rationale for describing the women in separate groups is not entirely clear.

Line 170, page 8: If the study is cross-sectional (measurements only conducted at a single time point for each woman), I would avoid using the word “cohort” to describe the women, as this implies they were followed up over time. Many people use the word more loosely, but its not strictly correct from an epidemiological perspective.

Line 177, page 8, onwards: In the results text the authors provide P values to indicate differences between PM+ and PM- women across a series of clinical parameters. I would prefer to see some indication of the actual difference in values within the text as well as the table because the p value by itself does not tell you anything about the magnitude (and therefore clinical significance) of the differences.

Table 1: Perhaps change “Percent women with anemia” to “Percentage of women with anemia" or "Anemia, n (%)” and “Percent LBW babies” to "Percentage LBW" or "Low birth weight, n (%)”. Also, "Percentage of women using ITNs or "ITN use, n (%)" and explain in the footnote or elsewhere how ITN use was defined.

Figure 1, 2, 3: Ensure consistent positioning of labels for median values, p values. Some median values are crossing the horizontal bars so it looks a bit messy.

Line 262-263: To my understanding, rapidly declining immunity in the absence of boosting has not been well characterized, more just inferred, so perhaps add references here or qualify the statement.

Line 285: Change “women having used” to “women who had used”

Reviewer #2: This paper reports immune response against VAR2CSA among Cameroonian pregnant women after a long-term implementation of intermittent preventive treatment (IPT). The strengths of the study are that it uses a full-length of VAR2CSA, the DBL5e domain of VAR2CSA; which is one of the most immunogenic antigen DBL5e and multi-analyte platform to assess the immune responses against VAR2CSA. The research team is highly skilled to properly address the research aims. The authors diagnose malaria on the population of study by using Giemsa-Wright stained thick and thin blood smears. In addition, they used thin films of peripheral blood to assess parasites species. Antibodies against VAR2CSA levels and frequency were characterized by a multi-analyte platform. Statistical analyses are adequately used and emerging data are interesting. The authors also report a significant adherence of malaria in pregnancy preventive measures including IPT in the region of study. Moreover, firstly, they find that a low number of women attending the clinic had placental malaria. Secondly, they demonstrate that more than 50% of pregnant women with placental malaria (PM) had antibodies against VAR2CSA at delivery. Interestingly, they also show that pregnant women taking 3 doses of ITP-SP displayed higher levels of hemoglobin and low risk of placental infections at delivery. The authors conclude that their data will help to better improve the design of future clinical trials in malaria endemic areas, and the efficacy VAR2CSA-based vaccines against MIP.

The outcomes of this study demonstrate the significant impact of IPT-SP on VAR2CSA antibodies levels and frequency in pregnant women within a unique malaria epidemiological setting as Cameroon. Although, this study makes an important contribution to the field, I still do have major comments that need to be addressed.

Major comments

Comment #1

One of the major weakness of this study is the lack of a full interpretation of the authors’ data. The authors scarcely explain the physiological aspect of their results and how, in a clear and definite way, these data can help in the surveillance of pregnant women health during MIP, the improvement of IPT-SP treatment and the development of a VAR2CSA-based vaccine.

Comment #2

Lines 45-46

Can the authors rewrite the following sentence?

It’s well established that pregnancy specific-infected erythrocytes (IE) express several variable surface antigens (VSA). VSA group is composed of many antigens families including PfEMP1, Stevor and Rifin. VAR2CSA is the best-characterized antigen from PfEMP1 family. Knock-out study has showed that VARCSA is the main antigen but not the unique antigen since, 15 to 20% of CSA-selected IE still bound to CSA after var2csa gene invalidation (Viebig et al., 2007). Moreover, some studies have identified different novel proteins with unknown functions. These proteins have structural motifs suggesting that they may potentially be expressed at the surface of IE and involved in malaria in pregnancy pathogenesis (Fried M et al., 2007; Francis SE et al., 2008; Bertin G et al., 2013).

Comment #3

Lines 130-131

Various studies have shown the importance of the use of qPCR to diagnose malaria since submicroscopic infections are underestimate in many African regions. The authors shall discuss this as a limitation of their study if they can provide qPCR data of their samples.

Comment #4

Lines 137-138

Can the authors describe with more clarity, the source of these recombinant proteins: insect cells, bacteria?

Comment #5

Lines 154

Please add the range of age for the three malaria-naïve Americans and 12 Cameroonian men living in the area of study. Also, do clarify how the cut-off was designed. It is still not very clear whether the 3 American and 12 Cameroonian men were all tested to design the cut-off.

Moreover, Plamodium (P) vivax transmission is growing in many sub-Sahara African regions. Have the authors assessed malaria infections or potential previous P. vivax exposure in Cameroonian men?

Comment #6

Lines 197-198

The authors report this: “For the 130 women, Ab levels to FV2 and DBL5 were higher in PM+ women than PM- women (p = 0.0064 and 0.0182, respectively) (Fig 1A and B)”. Whether these high levels of antibodies against FV2 and DBL5e are either marker of infections or protective antibodies against PM isn’t discuss anywhere in the manuscript.

Comment #7

Figure 1 (A and B): Please plot individual American and Cameroonian men serum/plasma reactivity against FV2 and DBL5e on the graph.

Comment #8

Line 222-223

Can the authors comment these data?

Minor comments

1- Figures are very fuzzy.

2- The ability to inhibit IE adhesion to CSA is the hallmark of efficient antibodies against VAR2CSA in MIP. The authors did not discuss whether antibodies they have identified in the population of study have functional activity.

3- Line 229: why is the word “about” written in bold?

6. PLOS authors have the option to publish the peer review history of their article (what does this mean?). If published, this will include your full peer review and any attached files.

Reviewer #1: Yes: Julia C Cutts

Reviewer #2: No

---

## [Author Response · Author response to Decision Letter 0]

17 Jul 2020

Dear Editor Luzia Helena Carvalho,

We thank you and the two reviewers for evaluating our manuscript: PONE-D-20-13928, entitled "Antibodies to full-length and the DBL5 domain of VAR2CSA in pregnant women after long-term implementation of intermittent preventive treatment in Etoudi, Cameroon".

Please find here in our responses to reviewer's 1 and 2 comments.

Reviewer #1

I enjoyed reading this well-written manuscript examining antibody responses to the pregnancy-specific malaria protein VAR2CSA in women in Cameroon. Understanding more about the impact of intermittent preventive treatment in pregnancy on population levels of antimalarial immunity is important for designing appropriate malaria interventions to achieve malaria elimination.

I have no major concerns with the scientific approach or conclusions drawn from the findings. I would recommend the following minor changes:

Query #1

Line 60, page 3: Consider changing “efficient” vaccine to “effective” vaccine, but then you use effective twice, so perhaps change to the following, or similar:

“The development of an effective vaccine against PM may offer a sustainable solution to protect mothers and their babies from malaria-related morbidity and mortality in endemic areas”.

Response #1. The sentence has been revised as suggested by the Reviewer.

Query #2

Line 138-139, page 7: The reference provided (Ref 24; Babakhanyan et al 2016) in turn refers to earlier papers for details on coupling of recombinant proteins to MagPix microspheres, so I would suggest including the reference to these earlier papers.

Response #2. References to the earlier papers (Fouda et al., 2006; Banakhanyan et al., 2014) have been added.

Query #3

Line 162, page 8: Change “unpair t-test” to “unpaired t-test”. Also provide a general statement, similar to what is included in footnote to table 1, on when t-tests were used and when Mann-Whitney Rank Sum were used (e.g. normal versus non-normal continuous data)

Response #3.In the Statistical analysis section, the text has been changed to read:“Differences between groups were compared using unpaired t-test for normal continuous data or Mann-Whitney Rank Sum test for non-normal continuous data, ….” 

Line 168-170, page 8: Upon initial reading it was not clear what was meant by “among whom 105 were randomly enrolled between June 2013 and February 2014”. Does this mean that a subset of 105 were randomly selected from the total of 130, or that 105 was the number of women who were enrolled between June 2013 and February 2014 (with the remaining 25 enrolled outside this period)?. I continued to read, and this was clarified to some extent at line 171, but the rationale for describing the women in separate groups is not entirely clear.

Response #4. The text has been changed for clarification. It now reads: “Initially,105 women were enrolled between June 2013 and February 2014, among whom 9 women had PM, providing an estimated prevalence of PM of 8.6% (9/105). Then, between May and June 2014, 25 women were enrolled to collect additional samples from PM+ women.”

Query #5

Line 170, page 8: If the study is cross-sectional (measurements only conducted at a single time point for each woman), I would avoid using the word “cohort” to describe the women, as this implies they were followed up over time. Many people use the word more loosely, but its not strictly correct from an epidemiological perspective.

Response #5. The word “cohort” has been deleted.

Query #6

Line 177, page 8, onwards: In the results text the authors provide P values to indicate differences between PM+ and PM- women across a series of clinical parameters. I would prefer to see some indication of the actual difference in values within the text as well as the table because the p value by itself does not tell you anything about the magnitude (and therefore clinical significance) of the differences.

Response #6. Values from Table 1 have been added to the text next to the corresponding p values as suggested by the reviewer (although this seems a little redundant). 

Query #7

Table 1: Perhaps change “Percent women with anemia” to “Percentage of women with anemia" or "Anemia, n (%)” and “Percent LBW babies” to "Percentage LBW" or "Low birth weight, n (%)”. Also, "Percentage of women using ITNs or "ITN use, n (%)" and explain in the footnote or elsewhere how ITN use was defined.

Response #7. The text in the Table 1 has been changed to “Percentage (%) of women with anemia," “Percentage (%) of LBW babies,” and "Percentage (%) of women using ITNs.” Additionally, the following text has been added to the Table legend: “** Percentage of women who reported using insecticide treated bed nets (ITNs).”

Query #8

Figure 1, 2, 3: Ensure consistent positioning of labels for median values, p values. Some median values are crossing the horizontal bars so it looks a bit messy.

Response #8.Median values have been removed from the figure since they are not necessary and the P values have been repositioned in Figures 1,2,3.

Query #9

Line 262-263: To my understanding, rapidly declining immunity in the absence of boosting has not been well characterized, more just inferred, so perhaps add references here or qualify the statement.

Response #9. The sentence has been rewritten as follow: “The transmission of malaria is decreasing in many parts of the world and Ab levels to P. falciparum antigens may decline in the absence of boosting.”

Query #10

Line 285: Change “women having used” to “women who had used”

Response #10. The change has been made.

Reviewer #2: 

This paper reports immune response against VAR2CSA among Cameroonian pregnant women after a long-term implementation of intermittent preventive treatment (IPT). The strengths of the study are that it uses a full-length of VAR2CSA, the DBL5e domain of VAR2CSA; which is one of the most immunogenic antigen DBL5e and multi-analyte platform to assess the immune responses against VAR2CSA. The research team is highly skilled to properly address the research aims. The authors diagnose malaria on the population of study by using Giemsa-Wright stained thick and thin blood smears. In addition, they used thin films of peripheral blood to assess parasites species. Antibodies against VAR2CSA levels and frequency were characterized by a multi-analyte platform. Statistical analyses are adequately used and emerging data are interesting. The authors also report a significant adherence of malaria in pregnancy preventive measures including IPT in the region of study. Moreover, firstly, they find that a low number of women attending the clinic had placental malaria. Secondly, they demonstrate that more than 50% of pregnant women with placental malaria (PM) had antibodies against VAR2CSA at delivery. Interestingly, they also show that pregnant women taking 3 doses of ITP-SP displayed higher levels of hemoglobin and low risk of placental infections at delivery. The authors conclude that their data will help to better improve the design of future clinical trials in malaria endemic areas, and the efficacy VAR2CSA-based vaccines against MIP.

The outcomes of this study demonstrate the significant impact of IPT-SP on VAR2CSA antibodies levels and frequency in pregnant women within a unique malaria epidemiological setting as Cameroon. Although, this study makes an important contribution to the field, I still do have major comments that need to be addressed.

Major comments

Comment #1

One of the major weakness of this study is the lack of a full interpretation of the authors’ data. The authors scarcely explain the physiological aspect of their results and how, in a clear and definite way, these data can help in the surveillance of pregnant women health during MIP, the improvement of IPT-SP treatment and the development of a VAR2CSA-based vaccine.

Response #1. The Discussion has been extensively revised to include a more biological and practical interpretation of the results. We now point out that many women, including multigravidae, in the Etoudi area of Yaoundé lack antibodies to VAR2CSA and may remain susceptible to placental malaria. Accordingly, all women, not just primigravidae, would benefit from a VAR2CSA vaccine. A statement was included that if clinicians and pregnant women followed the new WHO recommendation (SP at every prenatal visit) the number of PM cases could be reduced further.

 As for using pregnant women for surveillance of malaria, unlike in Mozambique (Fonseca et al. 2019), malaria transmission remainshigh in Cameroon with many children having clinical cases of malaria, so restricting surveillance to pregnant women whose infections are being controlled/ eliminated SP does not seem practical at this time. The most interesting finding of the study is that many women who have PM at delivery don’t have Ab to VAR2CSA, including many multigravidae. It is likely some of the multigravidaedid not become infected during previous pregnancies and produced a primary Ab response during the current pregnancy. The absence of Ab in low transmission areas suggests that VAR2CSA is not highly immunogenic and that Ab to other malarial antigens may help control parasitemia (Lloyd et al.,2018). These ideas and well as others (see below) are included in the revised manuscript.

Comment #2

Lines 45-46

Can the authors rewrite the following sentence?

It’s well established that pregnancy specific-infected erythrocytes (IE) express several variable surface antigens (VSA). VSA group is composed of many antigens families including PfEMP1, Stevor and Rifin. VAR2CSA is the best-characterized antigen from PfEMP1 family. Knock-out study has showed that VARCSA is the main antigen but not the unique antigen since, 15 to 20% of CSA-selected IE still bound to CSA after var2csa gene invalidation (Viebig et al., 2007). Moreover, some studies have identified different novel proteins with unknown functions. These proteins have structural motifs suggesting that they may potentially be expressed at the surface of IE and involved in malaria in pregnancy pathogenesis (Fried M et al., 2007; Francis SE et al., 2008; Bertin G et al., 2013).

Response #2. The sentence has been rewritten: “In pregnant women, Plasmodium falciparum-infected erythrocytes (IE) express an antigen, VAR2CSA, that participates in the binding of IE to chondroitin sulfate A (CSA) on the syncytiotrophoblast lining the intervillous space of the placenta [1, 2].” The revised statement states the function of VAR2CA, but doesn’t claim it is the only ligand involved. We hope this more accurately related the literature.

Comment #3

Lines 130-131

Various studies have shown the importance of the use of qPCR to diagnose malaria since submicroscopic infections are underestimate in many African regions. The authors shall discuss this as a limitation of their study if they can provide qPCR data of their samples.

Response #3. The Reviewer is correct that PCR was not used in this study to detect submicroscopic infections. We have stated this is a limitation in the current study in the Discussion.

Comment #4

Lines 137-138

Can the authors describe with more clarity, the source of these recombinant proteins: insect cells, bacteria?

Response #4. The following information has been added.“The recombinant proteins were from the FcR3 strain of P. falciparum and expressed in Baculovirus-transfected insect cells.”

Comment #5

Lines 154

Please add the range of age for the three malaria-naïve Americans and 12 Cameroonian men living in the area of study. Also, do clarify how the cut-off was designed. It is still not very clear whether the 3 American and 12 Cameroonian men were all tested to design the cut-off.

Moreover, Plamodium (P) vivax transmission is growing in many sub-Sahara African regions. Have the authors assessed malaria infections or potential previous P. vivax exposure in Cameroonian men?

Response #5. For clarity, the text has been changed to read: “Negative controls were plasma from 12 Cameroonian men, age 23 to 30 years of age, living in the study area. The cut-off value for seropositivity was determined based on the mean MFI + 2SD of the male samples.” FYI: The US controls were only used as traditional plate controls. Since data from US controls were not used to calculate the cut-off, the mention of US controls has been deleted.

As for the question about Pv, we are aware of the cross-reactivity of PvDBP and VAR2CSA (e.g., Mitran et al. 2019; Gnidehou et al., 2019). The male samples used in the study were not tested for Pv by PCR. The only information we have is that i) 20 years ago we tested over 200 individuals in Yaoundé for Pv by PCR and all were negative, ii) Pv has not been detected by microscopy in Yaoundé,iii) there is no clinical evidence of Pv infections, and iv) the few reported PCR-positive individuals for Pv in Cameroon resided in northern/west Cameroon (i.e., not in Yaoundé). Since the male controlsin the study had low MFI to full-length FV2, it seems highly unlikely that Pv had an impact on the current study. But it is something to keep in mind in future studies.

Comment #6

Lines 197-198

The authors report this: “For the 130 women, Ab levels to FV2 and DBL5 were higher in PM+ women than PM- women (p = 0.0064 and 0.0182, respectively) (Fig 1A and B)”. Whether these high levels of antibodies against FV2 and DBL5e are either marker of infections or protective antibodies against PM isn’t discuss anywhere in the manuscript.

Response #6. Lines 203 -204 in the original MS attempted to address this topic: “Thus, high Ab levels were associated with infection and many women lacked Ab to the VAR2CSA antigens.” A statement has been added to the Discussion stating that high Ab levels are a marker of infection, not protection.

Comment #7

Figure 1 (A and B): Please plot individual American and Cameroonian men serum/plasma reactivity against FV2 and DBL5e on the graph.

Response #7. Data for Cameroonian males has been included on Figure 1. Data for the 3 American samples were not included since they are no longer mentioned in the study. 

Comment #8

Line 222-223

Can the authors comment these data?

Response #8. Data showed there was no statistically-significant difference between Ab levels to FV2 in pauci- and multi-gravidae (MG) who had placental malaria (PM) (median: FV2 10,459 vs 21,078 MFI, p=0.10). However, the range of the data points is large, i.e., from 300 to 33,000 MFI, and there is no central tendency of the distribution.The most logical interpretationis that some MG women have NOT become infected during previous pregnancies due to low malaria transmission and the use of IPT-SP in previous pregnancies. Thus, although they were MG based on pregnancy status, some of the women produced only a primary Ab response during pregnancy. Thanks for pointing this out. We will include this information in the Discussion.

Minor comments

Quarry #1

 Figures are very fuzzy.

Response #1. The resolution of figures has been improved. They were converted to tiff files via PACEV2.

Quarry #2

2- The ability to inhibit IE adhesion to CSA is the hallmark of efficient antibodies against VAR2CSA in MIP. The authors did not discuss whether antibodies they have identified in the population of study have functional activity.

Response #2.Alas, functional Ab assays, neither inhibition of binding nor phagocytosis, can be conducted in the laboratory in Cameroon. The current study, including the multiplex assay was conducted at the Biotechnology Centre, Cameroon. Thus, we didn’t determine if any of the Ab to VAR2CSA detected are able to inhibit binding. Additional studies with a significantly large sample size would be required to assess the influence of IPT-SP on “level of functional Ab” and if functional Ab are associated with improved pregnancy outcomes. A paragraph has been added to the Discussion stating limitations of the study and future needs to do functional studies.

Quarry #3

3- Line 229: why is the word “about” written in bold?

Response #3. It was a mistake. It has been corrected

---

## [Decision Letter · Decision Letter 1]

31 Jul 2020

Antibodies to full-length and the DBL5 domain of VAR2CSA in pregnant women after long-term implementation of intermittent preventive treatment in Etoudi, Cameroon

PONE-D-20-13928R1

Dear Dr. DJONTU,

We’re pleased to inform you that your manuscript has been judged scientifically suitable for publication and will be formally accepted for publication once it meets all outstanding technical requirements.

Kind regards,

Luzia Helena Carvalho, Ph.D.

Academic Editor

PLOS ONE

Additional Editor Comments (optional):

Reviewers' comments:

Reviewer's Responses to Questions

**Comments to the Author**

1. If the authors have adequately addressed your comments raised in a previous round of review and you feel that this manuscript is now acceptable for publication, you may indicate that here to bypass the “Comments to the Author” section, enter your conflict of interest statement in the “Confidential to Editor” section, and submit your "Accept" recommendation.

Reviewer #1: All comments have been addressed

Reviewer #2: All comments have been addressed

2. Is the manuscript technically sound, and do the data support the conclusions?

Reviewer #1: Yes

Reviewer #2: Yes

3. Has the statistical analysis been performed appropriately and rigorously? 

Reviewer #1: Yes

Reviewer #2: Yes

4. Have the authors made all data underlying the findings in their manuscript fully available?

Reviewer #1: Yes

Reviewer #2: Yes

5. Is the manuscript presented in an intelligible fashion and written in standard English?

Reviewer #1: Yes

Reviewer #2: Yes

6. Review Comments to the Author

Reviewer #1: (No Response)

Reviewer #2: Thank you for addressing all my comments. The revised paper is well-written and addresses an important topic which is relevant for women health.

I trust that the revised manuscript is suitable for publication in the Journal of PLOS ONE.

Please find my comments related to the following subjects:

Dual publication: I have not seen any dual publication

Research ethics: the authors have followed all research ethics for the paper

Publication ethics: the paper fits publication ethics.

7. PLOS authors have the option to publish the peer review history of their article (what does this mean?). If published, this will include your full peer review and any attached files.

Reviewer #1: **Yes: **Julia Cutts

Reviewer #2: No

---

## [Editor Report · Acceptance letter]

5 Aug 2020

PONE-D-20-13928R1 

Antibodies to full-length and the DBL5 domain of VAR2CSA in pregnant women after long-term implementation of intermittent preventive treatment in Etoudi, Cameroon 

Dear Dr. DJONTU:

I'm pleased to inform you that your manuscript has been deemed suitable for publication in PLOS ONE. Congratulations! Your manuscript is now with our production department. 

Kind regards, 

on behalf of

Dr. Luzia Helena Carvalho 

Academic Editor

PLOS ONE